# Genome-Wide Identification and Expression Analysis of *TCP* Transcription Factors Responding to Multiple Stresses in *Arachis hypogaea* L.

**DOI:** 10.3390/ijms26031069

**Published:** 2025-01-26

**Authors:** Yanting Zhu, Sijie Niu, Jingyi Lin, Hua Yang, Xun Zhou, Siwei Wang, Xiaoyan Liu, Qiang Yang, Chong Zhang, Yuhui Zhuang, Tiecheng Cai, Weijian Zhuang, Hua Chen

**Affiliations:** 1Research Center of Leguminous Oil Plant Genetics and Systems Biology, College of Agriculture, Fujian Agriculture and Forestry University, Fuzhou 350002, China; yantingzhu1215@163.com (Y.Z.); s2792971695@163.com (S.N.); linjy0725@163.com (J.L.); 15121440955@163.com (H.Y.); 19180837179@163.com (X.Z.); 15159501851@163.com (S.W.); lxy15288528486@126.com (X.L.); fafuyq@163.com (Q.Y.); czhang@fafu.edu.cn (C.Z.); yuhuiz0902@163.com (Y.Z.); caitiecheng1027@163.com (T.C.); weijianz1@163.com (W.Z.); 2Key Laboratory of Fujian-Taiwan Crop Biological Breeding and Agriculture, Fujian Agriculture and Forestry University, Fuzhou 350002, China; 3Key Laboratory of Crop Genetics and Comprehensive Utilization, Ministry of Education, Fuzhou 350002, China

**Keywords:** peanut, *TCP* gene, genome-wide identification, bioinformatics, expression pattern analysis

## Abstract

The *TEOSINTE-BRANCHED1/CYCLOIDEA/PROLIFERATING-CELL-FACTOR* (*TCP*) gene family, a plant-specific transcription factor family, plays pivotal roles in various processes such as plant growth and development regulation, hormone crosstalk, and stress responses. However, a comprehensive genome-wide identification and characterization of the *TCP* gene family in peanut has yet to be fully elucidated. In this study, we conducted a genome-wide search and identified 51 *TCP* genes (designated as *AhTCPs*) in peanut, unevenly distributed across 17 chromosomes. These *AhTCPs* were phylogenetically classified into three subclasses: PCF, CIN, and CYC/TB1. Gene structure analysis of the *AhTCPs* revealed that most *AhTCPs* within the same subclade exhibited conserved motifs and domains, as well as similar gene structures. Cis-acting element analysis demonstrated that the *AhTCP* genes harbored numerous cis-acting elements associated with stress response, plant growth and development, plant hormone response, and light response. Intraspecific collinearity analysis unveiled significant collinear relationships among 32 pairs of these genes. Further collinear evolutionary analysis found that peanuts share 30 pairs, 24 pairs, 33 pairs, and 100 pairs of homologous genes with *A. duranensis*, *A. ipaensis*, *Arabidopsis thaliana*, and *Glycine max*, respectively. Moreover, we conducted a thorough analysis of the transcriptome expression profiles in peanuts across various tissues, under different hormone treatment conditions, in response to low- and high-calcium treatments, and under low-temperature and drought stress scenarios. The qRT-PCR results were in accordance with the transcriptome expression data. Collectively, these studies have established a solid theoretical foundation for further exploring the biological functions of the *TCP* gene family in peanuts, providing valuable insights into the regulatory mechanisms of plant growth, development, and stress responses.

## 1. Introduction

Transcription factors (TFs) play a crucial role in the regulation of gene expression. They regulate the transcription rate and expression level of genes by recognizing and binding specific sequences on DNA so as to participate in cell differentiation [1], proliferation [2], and adaptive response to the external environment [3]. Among them, the TEOSINTE BRANCHED 1, CYCLOIDEA, and PROLIFERATING CELL FACTOR (TCP) transcription factors, as a plant-specific transcription factor family, play an important role in many biological processes such as plant growth and development [4,5,6] and stress resistance [7,8]. The TCP protein is generally composed of 58–62 amino acids and contains a highly conserved basic helix–loop–helix (bHLH) structure. This structure enables TCP proteins to participate in DNA binding, protein–protein interactions, and nuclear localization of proteins [9]. TCP proteins are divided into two major classes according to the amino acid sequence similarity of the TCP domain: class I (PCF or TCP-P class) and class II (TCP-C class). Compared to class II, class I proteins have four amino acids missing from their basic domain [10,11]. TCP class II is further subdivided into the CINCINNATA (CIN) and CYC/TB1 subclades [11]. In addition, several class II TCP TFs have an arginine-rich motif (R domain) and an ECE motif (a glutamic acid–cysteine–glutamic acid stretch), with unknown functions of both domains [11]. Generally, class I TCP proteins can promote cell proliferation in leaves. However, class II TCP proteins can inhibit cell division, functioning as negative regulators of leaf growth and positive regulators of aging [6,12,13,14,15,16].

Much evidence shows that TCP TFs play diverse functions in multiple biological processes during plant growth and development, such as leaf morphogenesis and development [17,18], flower development [19,20], leaf senescence [21], seed germination [22], seed development [22,23], trichomes development [24,25], shoot development [26], branching [27], plant hormone signaling pathways [28], response to environmental stress, and plant immunity [29,30]. Recent evidence shows that TCP genes play important roles in cotton fiber development [31,32]. Interestingly, TCP TFs have an unexpected role in seed oil biosynthesis [33]. Furthermore, TCP from cucumber (*Cucumis sativus* L.) controls the identity, mobility, and development of tendrils [34,35]. The *TB1* gene in maize is the first gene found in the TCP family to regulate plant branch development. The mutant with deletion of *TB1* gene synthesis in maize showed a significant decline in apical dominance and a significant increase in the number of lateral branches and was unable to develop normal female inflorescence [36]. Similarly, the *OsTB1* gene in rice functions analogously to the *TB1* gene in maize, negatively regulating the development of lateral branches [4]. Furthermore, AtTCP5, AtTCP13, AtTCP17, and AtTCP24 in *Arabidopsis thaliana* are related to leaf development [12]. Additionally, CYCLOIDEA-like genes play a crucial role in controlling floral symmetry, floral orientation, and nectar guide patterning [37]. *TCP* genes are not only involved in the regulation of flower organ development but are also regulated by environmental signals. Different members of the TCP family in tea plants showed different responses to salt stress and low-temperature stress. The expressions of *TCP19*, *TCP20*, *TCP12*, and *TCP32* were upregulated after salt stress and low-temperature treatment, whereas the expressions of *TCP18* and *TCP30* were downregulated after low-temperature treatment [38]. These discoveries illustrate the importance of TCP TFs in plant growth and development as well as responding to multiple stresses.

In recent years, a large amount of TCP family members have been characterized in various plants with the deciphering of whole genome sequences, such as *Zea mays* [39], *Gossypium spp.* [40], *Solanum lycopersicum* [41], *Oryza sativa* [42], *Medicago sativa* [43], *Cymbidium goeringii* [44], *Camellia sinensis* [45], *Panicum virgatum* [46], *Prunus mume* [47], etc. Notably, significant progress has been achieved in the functional studies of *TCP* gene family members across multiple plant species. For instance, the TCP family transcription factor StAST1 interacts with FT-like proteins and StABL1 proteins to jointly regulate the maturation process in potatoes [48]. Additionally, the transcription factors *SlTCP24/29* are pivotal in regulating the development of compound leaves in tomatoes and facilitating the accumulation of cytokinins [49]. Furthermore, TCP transcription factors have been identified as negative regulators of trichome formation on the cotyledon epidermis of *Arabidopsis thaliana* [50].

However, the TCP family in peanut (*Arachis hypogaea* L.) has yet to be comprehensively described, and little is known about this gene family. Peanut is a major agronomic crop providing important sources of plant oil and proteins worldwide. However, peanut is easily affected by multiple biotic or abiotic stresses during the growth and development. It is of importance to identify the novel gene resources responding to the various stresses. Due to the diverse biological functions of TCP TFs during plant growth and development, we are interested in focusing on the comprehensive analysis of the *TCP* family genes in peanut. In recent years, the whole genome of cultivated peanut was deciphered [51], which provides a great opportunity to identify the TCP TFs in peanut.

In this study, 51 non-redundant TCP TFs were identified from peanut. Then, systematic analysis including chromosome location, phylogenetic relationships, gene structure, conserved motif, and cis-acting elements were analyzed. The expression profiling in different tissues, in different stages of pod development, in response to calcium deficiency, and in response to hormones and abiotic stresses was analyzed. Furthermore, the qRT-PCR analysis method was utilized to examine the expression patterns of the selected *TCP* genes under different abiotic stresses. This study provides valuable information for understanding the peanut TCP TFs and further facilitates functional characterization and molecular mechanism of TCP members in peanut growth and development.

## 2. Results

### 2.1. Identification and Characterization of TCP Family Members in Peanut

Based on the BlastP against Arabidopsis TCP protein sequences and HMMER search results, there were a total of 51 TCPs that contain the TCP domain in the peanut genome, named AhTCP1 to AhTCP51 based on their physical location on the chromosome (Appendix A). The 51 AhTCP proteins had an average length of 347 amino acids, ranging from 126 (AhTCP8) to 496 (AhTCP11). The average molecular weight (MW) was 37.69795 kDa (range 13.06654–53.16810 kDa). The isoelectric point (pI) ranges from 4.76 (AhTCP15) to 9.72 (AhTCP19), indicating a relatively high level of acidity. The grand average of hydropathicity (GRAVY) of all AhTCP proteins was negative, indicating that they were all hydrophilic proteins. Subcellular localization prediction showed that the majority of AhTCP proteins were located in the nucleus. In addition, AhTCP1, AhTCP14, AhTCP46, and AhTCP51 were predicted to be localized in the chloroplast, whereas a few AhTCP proteins were predicted to be present in the cytoskeleton, mitochondria, and cytoplasm (Appendix A).

### 2.2. Phylogenetic Analysis and Classification of AhTCP Genes

To analyze the phylogenetic relationships of AhTCP proteins, a phylogenetic tree was constructed based on the amino acid sequences of 51 AhTCP proteins, 24 AtTCP proteins, 54 GmTCP proteins, 10 AiTCP proteins, and 11 AdTCP proteins (Appendix A) using the maximum likelihood (ML) method by MEGA11 software (version 11.0.13, Tokyo Metropolitan University, Tokyo, Japan) (Figure 1a). Based on their homology with TCP proteins in other species, 51 AhTCP proteins were classified into two major branches and further divided into three clades, of which the PCF subfamily contained 26 AhTCP genes, the CIN subfamily contained 15 *AhTCP* genes, and the CYC/TB1 subfamily contained 10 *AhTCP* genes (Figure 1a). The PCF clade contained the most *AhTCP* genes, accounting for 50.98% of all *AhTCP* genes, followed by the CIN clade. The CYC/TB1 subfamily contained the fewest *AhTCP* genes, accounting for only 19.60%. The outcome resembles that observed in *Arabidopsis thaliana* and *Glycine max* (Figure 1b). Specifically, the PCF clade exhibits the highest gene count in both *Arabidopsis thaliana* (with 13 genes) and *Glycine max* (with 26 genes). Concurrently, *Arabidopsis thaliana* has eight genes belonging to the CIN subfamily, whereas soybean boasts nineteen genes within the same subfamily. Notably, the CYC/TB1 subfamily contains the least number of genes in both *Arabidopsis thaliana* and soybean.

### 2.3. Chromosome Localization and Collinearity Analysis of AhTCP Genes

Genomic analysis revealed that the 51 *AhTCP* genes were unevenly distributed within 17 out of 20 peanut chromosomes, with the exception of the chromosomes 2, 7, and 17 (Figure 2a). There were seven peanut *TCP* genes on chromosome 13, which distributed the most peanut TCP family members. In contrast, the chromosomes 4, 6, and 16 harbored the least, with only one each. Remarkably, chromosomes 3, 13, and 14 exhibited an enrichment in AhTCPs, with over 37.25% of TCP clusters on chromosomes 3, 13, and 14.

To investigate the duplication events of the *AhTCP* genes, a collinearity analysis of these identified *TCP* genes was conducted. There are a total of 32 pairs of segmentally duplicated genes in the TCP gene family of peanut, which are distributed on chromosomes 1, 3, 4, 5, 6, 8, 9, 10, 11, 12, 13, 14, 15, 16, 18, 19, and 20 (Figure 2b), indicating that *TCP* genes have gone through extensive duplication events. Notably, *AhTCP* genes are found within the CIN, PCF, and CYC/TB1 subclades, actively participating in these segmental duplication events. The above results suggested that the segmental duplication is the predominant duplication event for *AhTCP* genes and is the primary driver of the expansion of the TCP gene family of peanut.

To understand the divergence time of the 28 collinear gene pairs, the synonymous substitution rate was calculated, and the divergence time was estimated (Table 1). The divergence time ranged from 6.67 × 10^−5^ million years ago (MYA) for the gene pair *AhTCP5*:*AhTCP28* to 1.40 × 10^−8^ MYA for *AhTCP4*:*AhTCP9*. In recent years, most members of the same subfamily have undergone duplication events, including *AhTCP32* and *AhTCP2*, *AhTCP12* and *AhTCP41*, *AhTCP13* and *AhTCP42*, *AhTCP47* and *AhTCP19*, and *AhTCP14* and *AhTCP43*.

Following our detailed analysis, we found that the Ka/Ks values for 27 collinear gene pairs are below 1, which indicates that these genes have undergone purifying selection within peanuts. Nevertheless, the *AhTCP5* and *AhTCP28* gene pair stands out with a Ka/Ks value greater than 1, hinting at the possibility that this specific gene pair has undergone notable mutations, ultimately resulting in alterations to their encoded proteins (Table 1).

In order to clarify the evolutionary relationships of *TCP* genes among various species, a multi-collinearity analysis was conducted involving *A. ipaensis*, A. duranensis, *A. thaliana*, and *Glycine max* (Figure 3). The results revealed that a total of 30 syntenic relationships for *A. hypogaea* were identified in the genome of *A. duranensis*. Furthermore, 24 syntenic relationships involving *AhTCPs* were discovered in the genome of *A. ipaensis*. Additionally, there were 33 syntenic relationships found between *AhTCPs* and *AtTCPs*. When comparing the genomes, a greater number of collinear gene pairs was observed between *A. hypogaea* and *Glycine max* than when comparing *A. hypogaea* with *A. ipaensis*, *A. duranensis*, or *A. thaliana*. Specifically, 100 syntenic relationships for *A. hypogaea* were detected in the genome of *Glycine max*.

### 2.4. Cis-Acting Element Analysis of AhTCP Genes

In order to further predict the cis-acting elements contained in the promoter region of the *AhTCP* genes, the 2000 bp upstream region of the *AhTCP* genes was extracted for cis-acting element analysis. The results showed that the promoter regions of *AhTCP* gene family members contained a large number of cis-acting regulatory elements related to stress response, plant growth and development, hormone response, and light response, including stress response (STRE), anaerobic induction (ARE), MYC, etc.; AAGAA motif, O2-site, and other plant growth and development elements; plant hormone response-related elements such as ABA (ABRE), salicylic acid (TCA element), and methyl jasmonate (CGTCA and TGACG motif); and light (G-box), Box4, and other light response-related components (Figure 4).

### 2.5. The Conserved Domain, Motif, and Gene Structure Analysis

In order to better understand the structural characteristics of AhTCP protein sequences, the conserved protein sequences were analyzed. A total of 20 different conserved motifs (named motif 1–20) were identified (Figure 5b). The research findings reveal that, with the exception of *AhTCP38*, *AhTCP23*, *AhTCP1*, *AhTCP8*, *AhTCP46*, and *AhTCP26*, all remaining *AhTCP* genes harbor motif2 and motif3. This implies that motif2 and motif3 demonstrate a considerable level of conservation and are pivotal in the evolutionary trajectory of the members belonging to the *AhTCP* gene family. Notably, motif2, motif3, and motif17 are shared conserved motifs among both class I and class II subfamilies. In contrast, motif1, motif4, and motif7 are exclusively conserved motifs of the class I subfamily, while motif5, motif15, and motif16 are unique to the class II subfamily. Domain analysis further reveals that all 51 peanut TCP family members contain either TCP conserved domains or TCP superfamily conserved domains.

To understand the structural composition of *AhTCP* genes, the genomic DNA sequences of *AhTCP* genes including CDS, intron, and UTR were compared and analyzed. The results indicate that a significant proportion of *AhTCP* genes lack introns (21, accounting for 41.17%) and UTRs (20, representing 39.21%) (Figure 5a).

### 2.6. In Silico Expression Analysis of Peanut TCP Gene Family

To explore the possible function of AhTCPs during peanut growth and development, the expression profiles of AhTCPs in different tissues at different development stages were investigated using an RNA-seq dataset available at PGR database (http://peanutgr.fafu.edu.cn/index.php, accessed on 31 August 2024). The findings revealed that forty-seven AhTCPs were expressed in different tissues, and four *AhTCP* genes (*AhTCP1*, *AhTCP8*, *AhTCP28*, and *AhTCP29*) were not expressed in any of the tested tissues, including leaves, stems, stem tips, roots, root–stem junctions, root tips, root nodules, gynophores, pericarps, testas, embryos, or cotyledons (Figure 6a). Among them, 23 *AhTCPs* were expressed in all the tissues. Notably, *AhTCP5*, *AhTCP15*, and *AhTCP44* were specifically expressed in the embryo, and *AhTCP14*, *AhTCP17*, *AhTCP27*, *AhTCP43*, and *AhTCP49* were highly expressed in the embryo and testa, indicating that these genes could play important roles during seed development. *AhTCP38* is notably expressed in the gynophore, implying that *AhTCP38* might regulate the characteristics of peanut underground fruiting. Furthermore, a cluster of *AhTCPs* had a high expression in the leaves, and these genes might be involved in the leaf development and morphogenesis.

Given the pivotal role of plant hormones in regulating plant growth, an analysis of *AhTCP* gene expression patterns induced by various exogenous plant hormones was conducted using RNA-seq data. The findings revealed that multiple *TCP* genes exhibited either upregulation or downregulation in response to different exogenous plant hormones. Notably, *AhTCP20*, *AhTCP19*, *AhTCP47*, *AhTCP6*, *AhTCP30*, *AhTCP37*, *AhTCP7*, *AhTCP31*, *AhTCP24*, *AhTCP16*, and *AhTCP45* were significantly induced by the salicylic acid (SA). Conversely, *AhTCP3*, *AhTCP12*, *AhTCP38*, *AhTCP2*, *AhTCP51*, *AhTCP39*, *AhTCP9*, *AhTCP32*, and *AhTCP43* underwent downregulation under the influence of abscisic acid (ABA), SA, brassinolide, paclobutrazol, and ethephon. Additionally, genes including *AhTCP14*, *AhTCP22*, *AhTCP11*, *AhTCP34*, *AhTCP10*, *AhTCP35*, *AhTCP33*, *AhTCP4*, and *AhTCP27* were also downregulated by ABA, brassinolide, paclobutrazol, and ethephon (Figure 6b).

Transcriptome expression data were also employed to investigate the expression levels of *AhTCPs* under varying abiotic stress conditions including drought and low temperature (4 °C) to evaluate their respective stress responses. The expression levels of *AhTCP26*, *AhTCP43*, *AhTCP14*, *AhTCP16*, *AhTCP17*, and *AhTCP22* were observed to increase under drought treatment, in contrast to normal irrigation conditions. Conversely, the expression levels of *AhTCP6*, *AhTCP30*, *AhTCP32*, *AhTCP39*, *AhTCP2*, *AhTCP9*, *AhTCP33*, *AhTCP24*, *AhTCP51*, *AhTCP3*, and *AhTCP4* were found to decrease under drought treatment compared to normal irrigation. During low-temperature treatment, the expression levels of *AhTCP45*, *AhTCP43*, *AhTCP14*, *AhTCP16*, *AhTCP38*, *AhTCP41*, *AhTCP34*, *AhTCP21*, *AhTCP50*, *AhTCP2*, *AhTCP9*, and *AhTCP33* were observed to elevate in comparison to their levels during room-temperature treatment. In contrast, *AhTCP7*, *AhTCP25*, *AhTCP31*, *AhTCP6*, *AhTCP30*, and *AhTCP26* showed reduced expression levels during low-temperature treatment when compared to room-temperature treatment (Figure 6c).

The developmental process of peanut embryos is highly susceptible to variations in various soil nutrient elements, especially the calcium level in the soil. Peanut produces abortive embryos in calcium-deficient soil [52]. Through comprehensive analysis of transcriptome data obtained from various stages of peanut embryo development in both low- and high-calcium soil environments, we investigated the expression dynamics of the *AhTCP* gene family in early peanut embryos in response to these calcium conditions [52]. Our findings revealed that, with the exception of a handful of genes that remained unaffected by calcium treatment, the majority of *AhTCP* genes demonstrated a marked upregulation under low-calcium conditions and a pronounced downregulation under high-calcium conditions (Figure 6d). This revelation leads us to hypothesize that these *TCP* genes hold a pivotal position in regulating the developmental trajectory of peanut embryos.

### 2.7. Expression of Selected AhTCP Genes Under Cold, ABA, and SA Treatments

Based on transcriptome data analysis, it was found that members of the *AhTCP* gene family exhibited different response patterns under abiotic stress conditions (Figure 6b,c). To validate this finding, the selected genes from the *AhTCP* family that showed significant upregulation or downregulation in expression levels under treatments with salicylic acid (SA), abscisic acid (ABA), and low-temperature stress were further tested and confirmed using quantitative real-time PCR (qRT-PCR) technology. The experimental findings demonstrated a variety of expression trends for *TCP* genes in peanuts within the 0 to 24 h period following abiotic stress treatments. Under cold stress, *AhTCPs* exhibit upregulation or downregulation of expression. For instance, the expression of *AhTCP33* gradually increased after cold treatment, whereas the expression of *AhTCP7* was lower at 6 h, 12 h, 18 h, and 24 h post-stress compared to non-stressed controls (Figure 7a). Similarly, under hormonal stress, *AhTCP* genes displayed specific response patterns. For example, under SA treatment, the expression levels of *AhTCP6*, *AhTCP31*, and *AhTCP36* initially increased and then gradually decreased. Likewise, under ABA treatment, the expression of *AhTCP50* also underwent a process of initial increase followed by a decrease (Figure 7b,c). Despite variations in the specific expression levels of these genes at different time points, the overall expression patterns obtained through qRT-PCR were consistent with the expression trends observed in transcriptome data (Figure 6b,c). These results not only reveal the complex response patterns of *AhTCP* gene family members under different stress conditions but also further confirm the accuracy and reliability of the transcriptome expression data.

## 3. Discussion

The *TCP* gene plays an important role in secondary metabolite accumulation, growth and development regulation, and hormone response [28,53,54,55]. TCP protein is involved in the regulation of seed germination, flowering, flower organ development, branch growth, and other processes but also through the interaction with other factors and the regulation of different hormone pathways [28,53,54,55]. Recently, with the development of high-throughput sequencing, *TCP* gene family members have been identified in multiple plant species, such as *Arabidopsis* [56], *Medicago sativa* [43], *Dendrobium nobile* [57], *Camellia sinensis* [45], *Cymbidium goeringii* [44], and *Prunus mume* [47]. Peanut is an important oil crop. However, there are no reports on *AhTCP* gene family members at the genome-wide level. Therefore, the *AhTCP* genes were systematically analyzed using published peanut genome-wide information and bioinformatic techniques in this study.

In this study, 51 *AhTCP* gene family members were identified from the peanut whole genome level (Appendix A). Compared with other crops such as *Medicago sativa* (71 *MsTCP* genes) [43], *Panax ginseng* (78 *PgTCP* genes) [58], and *Triticum aestivum* (66 *TaTCP* genes) [59], the number of *AhTCP* gene family members was on the high side and significantly more than that of *Arabidopsis thaliana* (24 *AtTCP* genes) [42], Solanum lycopersicum (21 *SlTCP* genes) [41], Senna tora (24 *StTCP* genes) [60], and *Camellia sinensis* (37 *CsTCP* genes) [45], which demonstrated that the number of *TCP* genes was closely related to the genome size and gene expansion degree among species. Based on the evolutionary relationship analysis results, the 51 *AhTCP* genes further were classified into three subclasses: PCF, CIN, and CYC/TB1 (Figure 1a), which was similar to the results of previous studies on *TCP* transcription factors [36,43]. The genes with similar motifs may have been evolutionarily conserved and perform the same functions. In this study, each subclade exhibited unique conserved motifs in peanut, implying that *AhTCP* genes belonging to different subclades may perform different functions (Figure 5b). For instance, motif1 is exclusively found in the PCF subfamily, whereas motif 15 is distinctly characteristic of class II CIN proteins, and motif13 is uniquely associated with class II CYC/TB1 proteins (Figure 5b). Additionally, our observations reveal that the types, quantities, and arrangements of conserved motifs within proteins encoded by members of the same subfamily show a remarkable degree of consistency (Figure 5b). This consistency leads us to hypothesize that it may serve as a crucial determinant of the functional distinctions among these three subfamilies [61].

Here, 51 *AhTCP* genes were found to be unevenly distributed on 17 chromosomes in the peanut genome (Figure 2a), which may be due to the fact that different chromosomes might have undergone different recombination, mutation, and selection events during evolution. It has been proved that introns affect gene expression [62] and that intronless genes can produce more rapid protein [63], so studying the structure of the *AhTCP* gene is beneficial to further reveal its function. We found most *AhTCP* genes (21/51, 41.17%) do not have introns (Figure 5a), implying the transcription times are shorter than those containing introns. Studies have demonstrated that in the plant species *Arabidopsis thaliana* and rice, genes belonging to specific gene families that are devoid of introns or exhibit low intron content are more prone to participating in physiological mechanisms for withstanding drought and salt stress in comparison to other genes [64]. Notably, these intronless subfamily genes are predominantly enriched in the biological process of ‘transcriptional regulation’ which operates within drought-responsive pathways [65], hinting at their pivotal roles in the adaptive responses to both drought and salt stresses. However, studies have shown that, somewhat counterintuitively, in scenarios of cellular nutrient deprivation, the presence of introns can actually stimulate cell growth. This occurs because introns have the ability to augment the repression of specific ribosomal protein genes situated downstream of the nutrient signaling cascades TORC1 and PKA, ultimately aiding the cell in more effectively managing starvation conditions [66]. It was predicted that an intron loss event may occur during gene evolution [67]. However, what is the role of the intron loss or the different number of introns retained of *AhTCP* genes requires further studies.

Genome collinearity and syntenic relationships are significant evolutionary phenomena that contribute to our understanding of genome duplication and neofunctionalization [68]. The analysis of collinearity has unveiled pivotal syntenic connections between *AhTCPs* and species such as *Arabidopsis thaliana*, *Glycine max*, *Arachis duranensis*, and *Arachis ipaensis*.

Predictive analysis of promoter cis-acting elements is an important tool for understanding gene function and its potential regulatory mechanisms. Predictive analysis of promoter cis-acting elements revealed that *AhTCP* contains a large number of cis-acting elements related to stress response, plant growth and development, plant hormone reactivity, and light response (Figure 4), suggesting that the *AhTCP* gene family may be involved in relevant regulatory pathways to affect peanut growth and development and stress response. However, the types and numbers of cis-acting elements varied in different *AhTCP* subfamily members. This result indicates that different *AhTCP* subfamily members might have functional differentiation and perform diverse functions under different conditions during peanut growth and development.

Previous studies have demonstrated that the *TCP* gene family plays a pivotal regulatory role in the cold and drought stress response across various plant species, earning it recognition as a valuable genetic resource for molecular drought-resistant breeding programs. Specifically, in cassava, more than 20% of *TCP* family members have been shown to respond to drought and cold stress treatments [69]. When the *CnTCP4* gene is overexpressed in Chrysanthemum, it leads to increased sensitivity to cold stress [70]. In rice, the *OsTCP19* gene undergoes significant upregulation under conditions of salt and drought stress [7], and overexpressing this gene in Arabidopsis results in transgenic plants that exhibit enhanced tolerance to abiotic stress [71]. Furthermore, Arabidopsis plants overexpressing *CsTCP5* and *CsTCP18* display superior survival rates under drought stress conditions [29], and similarly, transgenic Arabidopsis plants overexpressing *GbTCP4* also demonstrate increased survival rates under drought stress [72]. These observations further emphasize the important role that the *TCP* gene family plays in a plant’s response to cold and drought stress. To uncover *AhTCP* genes in peanut that possess the capacity to respond to and regulate drought stress, this study integrated bioinformatic analysis techniques with transcriptome data. By comparing the identified *AhTCP* genes against known transcriptome data, we discovered that the *AhTCP22* gene is uniquely expressed in leaf tissues (Figure 6a), with its promoter region abundant in diverse light-responsive elements (Figure 4). Notably, under drought conditions, the expression level of the *AhTCP22* gene increases substantially (Figure 6c). This characteristic hints at the potential involvement of the *AhTCP22* gene in the photosynthesis process [73], enhancing the plant’s water use efficiency and consequently bolstering its resilience to water stress [74].

Calcium is indispensable for the growth and development of peanuts. A deficiency of calcium in the soil can severely impair seed development, leading to unfilled pods and consequently resulting in decreased peanut yield and inferior quality [75]. Peanuts cultivated in calcium-depleted soil frequently suffer from high embryonic abortion rates and produce empty or incompletely filled pods, causing substantial economic losses to the global peanut industry, especially in tropical and subtropical regions [76]. In addressing this challenge, research has revealed that under conditions of low or high calcium, most *AhTCP* genes demonstrate different levels of upregulation or downregulation in their expression patterns (Figure 6d). These genes represent promising candidates for exploring how peanuts adapt to high- or low-calcium stress, thereby laying the groundwork for molecular breeding efforts aimed at developing peanut varieties that are resilient to low-calcium conditions while maintaining high yield and quality attributes.

Building upon the insights gained from this study, a comprehensive exploration of the peanut *TCP* gene family at the whole genome scale will greatly advance the identification of genes involved in peanut growth, development, and stress resistance. This endeavor will not only enrich our comprehension of peanut growth mechanisms and stress response mechanisms but also establish a robust theoretical and practical framework for developing new peanut varieties with accelerated growth rates and heightened stress tolerance through genetic engineering methodologies.

## 4. Materials and Methods

### 4.1. Identification and Sequence Analysis of Putative AhTCPs

The cultivated peanut genome was retrieved from the Peanut Genome Resource database (PGR) (http://peanutgr.fafu.edu.cn/index.php, accessed on 25 July 2024). The *TCP* genes in the diploid progenitors, specifically *A. duranensis* and *A. ipaensis*, were searched within the PeanutBase database (https://www.peanutbase.org/, accessed on 25 July 2024) [77]. Additionally, the *TCP* gene sequence of soybean was sourced from the NCBI (https://www.ncbi.nlm.nih.gov/, accessed on 25 July 2024). Meanwhile, all the AtTCP protein sequences were acquired from the TAIR website (https://www.arabidopsis.org/, accessed on 25 July 2024). The Hidden Markov Model (HMM) file of the conserved TCP domain (PF03634) was obtained from the Pfam database. Subsequently, HMMER 3.0 (http://hmmer.janelia.org/, accessed on 25 July 2024) was used to search AhTCPs with the e-value set to 1e-5 to obtain the candidate AhTCPs. All candidate TCP sequences were put in InterproScan [78] to confirm the presence of the conserved TCP domain, and the sequences without the TCP domain were manually eliminated. Then, the candidate AhTCP TFs were manually inspected with the Conserved Domain Database (CDD) (https://www.ncbi.nlm.nih.gov/Structure/cdd/cdd.shtml, accessed on 25 July 2024) and the simple modular architecture research tool (SMART) (http://smart.embl-heidelberg.de/, accessed on 25 July 2024) databases to confirm the presence of the conserved TCP domain. The candidate AhTCP proteins containing the TCP domain were obtained and used for further analysis. The physicochemical properties including the molecular weights (MWs), isoelectric points (pIs), and grand average of hydropathicity (GRAVY) of AhTCPs were estimated by the online tools ProtParam in ExPASy (http://web.expasy.org/protparam/, accessed on 25 July 2024). In addition, the subcellular localization of AhTCP proteins was predicted by the WoLF PSORT online website (https://wolfpsort.hgc.jp/, accessed on 25 July 2024).

### 4.2. Chromosomal Location

The chromosome location information of *AhTCP* genes were extracted from the GFF3 files of the peanut genome [51] and visualized by Gene Location Visualize from GTF/GFF module in TBtools software (version 1.0 (11.0.16), South China Agricultural University, Guangzhou, China) [79].

### 4.3. Collinearity Analysis of Peanut TCP Gene

The BLASTP program was used to identify homologous *TCP* genes in peanuts, and the E-value threshold was set to <10^−5^. The collinearity of *TCP* genes in peanut was analyzed using MCScanX default parameters. The resulting *TCP* collinear gene pairs were visualized using TBtools. To determine the synonymous (Ks) and non-synonymous (Ka) substitution rates, the simple Ka/Ks calculator available in TBtools software (version 1.0 (11.0.16)) was employed. The divergence time for duplicated gene pairs was calculated using the formula ‘t = Ks/2r’, where ‘r’ represents the neutral substitution rate, set at 8.12 × 10^−9^ [77]. The collinearity between peanut and *Arabidopsis thaliana*, *Glycine max*, *Arachis duranensis*, and *Arachis ipaensis* was analyzed by MCScanX program.

### 4.4. Gene Structure, Conserved Motif, and Phylogenetic Analysis of AhTCP Proteins

The structure of the peanut *TCP* gene, including intron, CDS, and UTR, was extracted from the peanut genome annotation file [51], and TBtools v2.119 [79] was used for visualization analysis. The conserved motifs of AhTCPs were identified using MEME (https://meme-suite.org/meme, accessed on 26 July 2024) (http://www.OMIcsclass.com/article/67, accessed on 26 July 2024) [80], where the maximum number of motifs was set to 20, the minimum width of motifs to 6, and the maximum width of motifs to 50.

Evolutionary phylogenetic analysis was conducted to assess the relationships between AhTCP and TCP proteins from *A. duranensis*, *A. ipaensis*, *Arabidopsis thaliana*, and *Glycine max*. Multiple sequence alignments of the amino acid sequences of these TCP proteins across different species were analyzed using ClustalX 2.1 [81]. A phylogenetic tree was constructed using the MEGA11.0 software (version 11.0.13) maximum likelihood (ML) method with the parameter Bootstrap value 1000 [82]. Then, iTOL (https://itol.embl.de/, accessed on 27 July 2024) was used to visualize and beautify the phylogenetic tree.

### 4.5. Cis-Acting Element Analysis of AhTCP Genes

TBtools software (version 1.0 (11.0.16)) was used to extract the nucleotide sequences 2000 bp region upstream of the start codon of the each *AhTCP* gene [79]. The cis-acting elements of the *AhTCP* gene family were predicted and analyzed using PlantCARE (http://www.dna.affrc.go.jp, accessed on 30 July 2024) online website. Subsequently, the Basic Biosequence View and HeatMap modules in TBtools were employed to visualize and summarize the cis-elements associated with stress response, plant growth and development, plant hormone reactivity, and light response.

### 4.6. Expression Analysis of the AhTCPs in Different Tissues, Hormone Conditions, and Diverse Stresses

The transcriptome data of *AhTCP* genes were download from the PGR database (http://peanutgr.fafu.edu.cn, accessed on 31 August 2024). TBtools-II was then used to reprocess the data and create heat maps for visualization, which normalized the data.

The RNA-seq data including various organs from the root, root-tip, stem, stem-tip, cotyledons, leaf, pericarp, testa, embryos, and the leaves at the 4-leaf-stage seedlings under low temperature (4 °C) and 8-leaf-stage seedlings under drought (no irrigation) condition were obtained from the PGR website and used to analyze the *AhTCP* expression in different tissues, organs, or responding to abiotic stresses. Additionally, the sequence data of peanut leaves under different exogenous plant hormone treatments including abscisic acid (ABA), salicylic acid (SA), brassinolide (BR), paclobutrazol (PAC), and ethylene (ET) were also downloaded from the PGR website. For the data, at the 7-leaf stage, peanut plants were sprayed with the following solutions on the underside of leaves: 10 µg/mL ABA, 3 mM SA, 0.1 mg/L BR, 150 mg/L PAC, and 1 mg/mL ETH or distilled water (mock).

The RNA-seq data (BioProject ID: PRJNA470988) including samples from embryos under calcium deficiency and sufficiency conditions at the stage of 15, 20, and 30 DAP were used to analyze the *AhTCP* expression responding to the calcium deficiency.

### 4.7. Stress Treatments and qRT-PCR Analysis

The peanut variety Minhua 6 (M-6) was selected and planted in plastic pots, each measuring 46 square centimeters and filled with nutrient soil, for cultivation in a greenhouse. The greenhouse was maintained at a constant temperature of 26 °C, with a daily photoperiod of 16 h of light and 8 h of darkness. Throughout the peanut’s growth cycle, the plants were watered with tap water every four days until they reached the four-leaf stage. At this point, the plants were sprayed with abscisic acid (ABA) at a concentration of 10 µg/mL and salicylic acid (SA) at a concentration of 3 Mmol and were subjected to low-temperature (4 °C) treatment. At 6, 12, 18, and 24 h post-treatment, peanut leaves were harvested as samples, with each time point being sampled in triplicate for consistency. The untreated leaf samples collected at 0 h served as the control group (CK) for comparison. RNA extraction was performed using the RNAprep Pure Plant Plus Kit (for Polysaccharides and Polyphenolic-rich samples) from Tiangen Biochem Technology (Beijing) Co., Ltd., Beijing, China. Subsequently, cDNA was synthesized using the HiScript II 1st Strand cDNA Synthesis Kit (+gDNA wiper) from Vazyme Biotech Co., Ltd. (Nanjing, China). The specific primers were designed using BioXM 2.7.1 software (version 2.1.7, Nanjing Agricultural University, Nanjing, China) (Appendix A). Real-time PCR for the relative expression level of the selected genes was performed, and *Actin* was used as an internal reference gene following the cycling program: 95 °C (30 s), 95 °C (10 s), and 60 °C (30 s) for 40 cycles [83]. All reactions were performed on an QuantStudio™ 3 system in triplicate. The relative expression levels of the selected genes were calculated using the 2^−ΔΔCT^ method [84]. GraphPad Prism 7.0 software (version 8.0.2, GraphPad Software, California, America) [85] was used for graph plotting. Compare the differences between control values and experimental values using the ANOVA method in GraphPad Prism 7.0 software [85].

## 5. Conclusions

In conclusion, we comprehensively investigated the *AhTCP* genes at the genome-wide level. The physicochemical properties, conserved structural domains and motif patterns, and genetic evolution were analyzed by the bioinformatic approach. In addition, the expression levels in different tissues and under different hormone or stress treatments were analyzed using RNA-seq. It was found that *AhTCP22* exhibits high expression levels in leaves, and its promoter region contains various light-responsive elements. Furthermore, under drought stress, the expression level of the *AhTCP22* gene increases significantly, suggesting that this gene can serve as a candidate for studying drought resistance mechanisms. Additionally, qRT-PCR was employed to assess the expression level changes for some *AhTCPs* in response to cold and hormone treatments. For instance, *AhTCP50* expression is upregulated under both low-temperature and ABA stress, while *AhTCP7* expression decreases under low-temperature stress but increases under SA stress. These findings indicate that *AhTCP50* and *AhTCP7* can be considered as candidate genes in the study of abiotic stress mechanisms. Our results provided a basis for further identifying the biological functions of *AhTCP* genes during peanut growth and development and responding to different stresses.

## Figures and Tables

**Figure 1 ijms-26-01069-f001:**
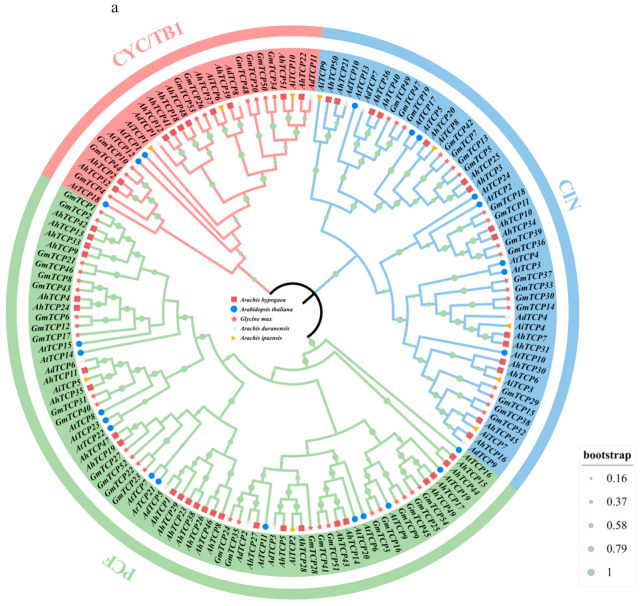
(**a**) Phylogenetic relationships among TCP proteins in *Arachis hypogaea* (AhTCP), *Arabidopsis thaliana* (AtTCP), *Glycine max* (GmTCP), *Arachis duranensis* (AdTCP), and *Arachis ipaensis* (AiTCP). The phylogenetic tree was constructed using the maximum likelihood (ML) method based on 150 full-length protein sequences from 51 AhTCPs, 24 AtTCPs, 54 GmTCPs, 11 AdTCPs, and 10 AiTCPs. The green, pink, and blue rings indicate PCF, CYC/TB1, and CIN subclades, respectively. (**b**) The ratios of TCP genes in *Arachis hypogaea*, *Arabidopsis thaliana*, and *Glycine max* among the three subfamilies of CIN, CYC/TB1, and PCF.

**Figure 2 ijms-26-01069-f002:**
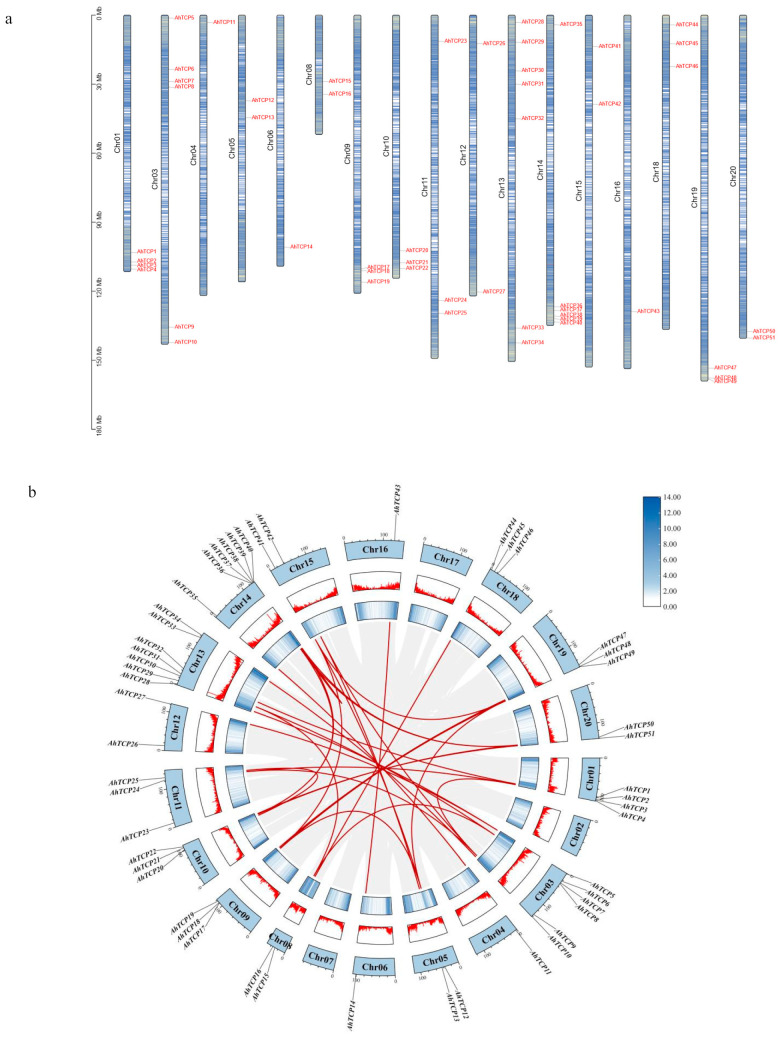
(**a**) Chromosomal distribution of *AhTCP* genes. The scale on left side indicates the chromosome length. (**b**) Schematic diagram of gene duplication in *AhTCPs*. Gray lines show the collinear regions of *Arachis hypogaea* with other species. Red lines indicate the duplicated gene pairs.

**Figure 3 ijms-26-01069-f003:**
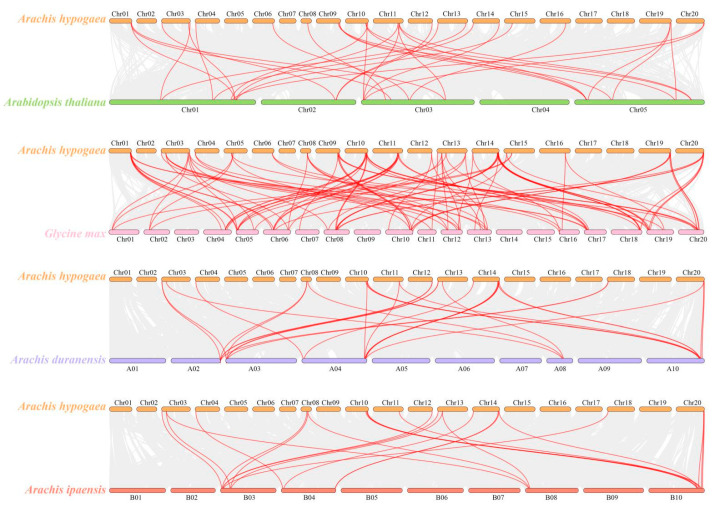
Synteny analysis of *Arachis hypogaea*, *Arabidopsis thaliana*, *Glycine max*, *Arachis duranensis*, and *Arachis ipaensis*. Gray lines show the collinear regions of *Arachis hypogaea* with other species. Red lines indicate the duplicated gene pairs.

**Figure 4 ijms-26-01069-f004:**
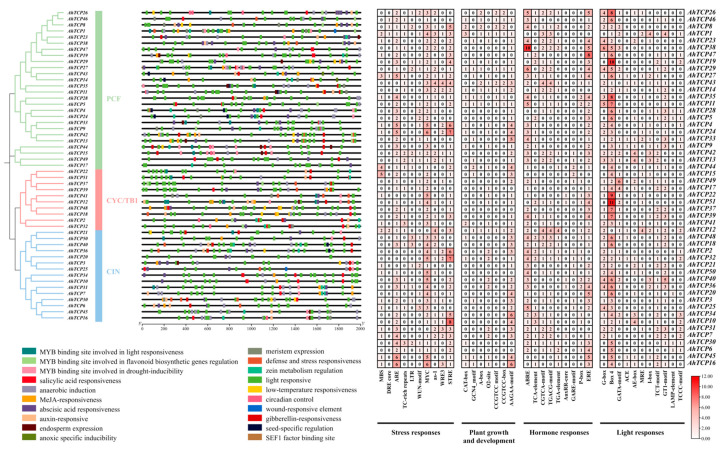
Cis-acting elements in promoters of AhTCP family members.

**Figure 5 ijms-26-01069-f005:**
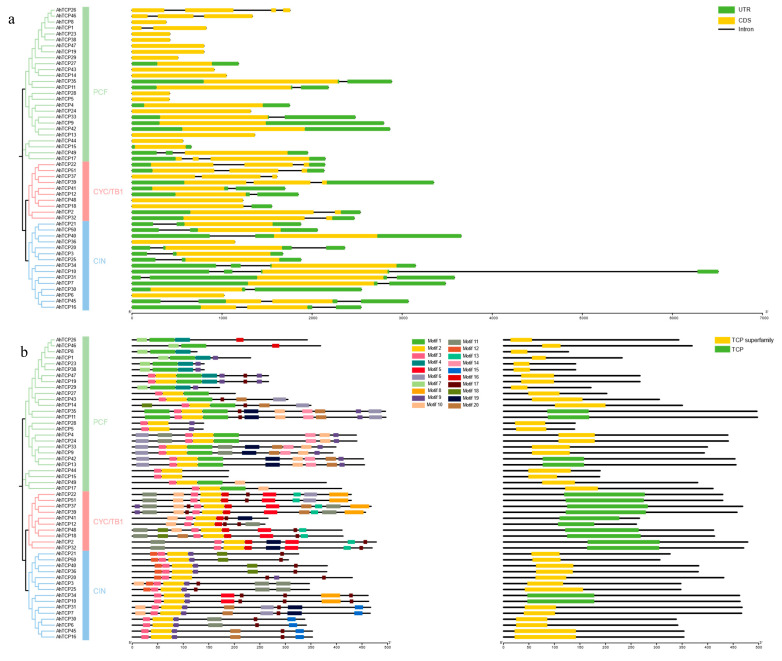
Structural analysis of *AhTCP* genes in peanut. (**a**) Gene structure. (**b**) The conserved motifs in AhTCP proteins.

**Figure 6 ijms-26-01069-f006:**
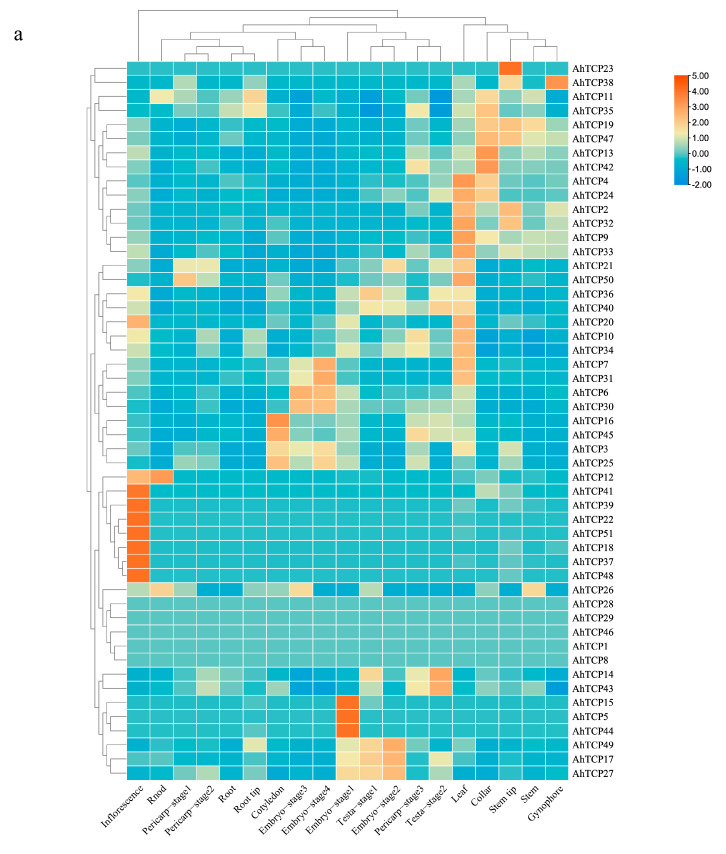
Transcriptome expression of AhTCPs (**a**) in various tissues, (**b**) under different hormone treatments, (**c**) in stress conditions, and (**d**) under low-calcium and high-calcium treatments.

**Figure 7 ijms-26-01069-f007:**
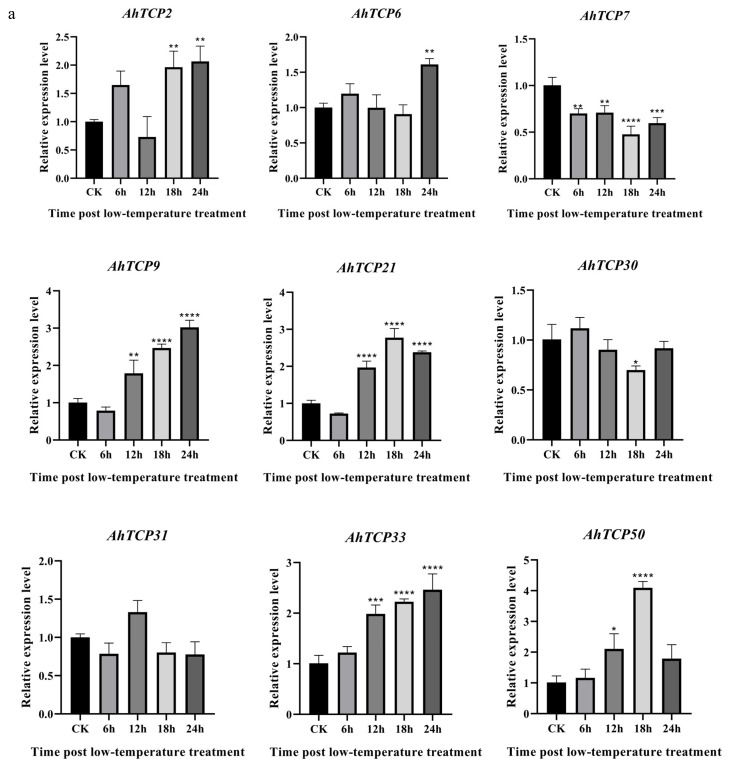
Relative expression analysis of *AhTCP* genes in peanut leaf in response to cold stress (**a**), SA (**b**), and ABA (**c**) treatment. CK denotes the control samples, whereas 6 h, 12 h, 18 h, and 24 h signify the time points in hours following the application of stress treatments, including low temperature, SA (salicylic acid), and ABA (abscisic acid). The data were analyzed using the 2^−△△CT^ method, and statistical significance was ascertained through ANOVA (* *p*  <  0.05, ** *p*  <  0.01, *** *p*  <  0.001, **** *p*  <  0.0001).

**Table 1 ijms-26-01069-t001:** Calculation of Ka/Ks for duplicated gene pairs.

Seq_1	Seq_2	Ka	Ks	Ka/Ks	Ks/2r (MYA)
AhTCP4	AhTCP9	0.352897198	2.2657562	0.1557525	139,517,004.99
AhTCP4	AhTCP24	0	0.0265001	0.0000000	1,631,781.68
AhTCP2	AhTCP32	0.014846228	0.0383025	0.3876048	2,358,527.80
AhTCP6	AhTCP16	0.285208236	1.9666324	0.1450237	121,098,056.02
AhTCP9	AhTCP24	0.334979524	2.1683105	0.1544887	133,516,659.16
AhTCP5	AhTCP27	0.489917965	NaN	NaN	NaN
AhTCP10	AhTCP34	0.003780726	0.0298712	0.1265676	1,839,359.90
AhTCP6	AhTCP30	0.009008151	0.0377587	0.2385718	2,325,040.73
AhTCP5	AhTCP28	0.016047684	0.0108207	1.4830477	666,302.14
AhTCP9	AhTCP42	0.262350405	1.8892023	0.1388683	116,330,190.81
AhTCP11	AhTCP35	0.009853829	0.0494365	0.1993230	3,044,119.23
AhTCP12	AhTCP18	0.404993521	1.1462073	0.3533336	70,579,265.77
AhTCP13	AhTCP42	0.005808823	0.0586459	0.0990491	3,611,199.17
AhTCP12	AhTCP41	0.003335655	0.0361334	0.0923149	2,224,964.91
AhTCP12	AhTCP48	0.390690818	1.0952004	0.3567300	67,438,450.47
AhTCP14	AhTCP43	0.004258351	0.0297806	0.1429908	1,833,780.00
AhTCP16	AhTCP30	0.265356041	2.2656185	0.1171230	139,508,527.38
AhTCP15	AhTCP44	0.011964361	0.0485017	0.2466790	2,986,560.63
AhTCP18	AhTCP41	0.518842675	1.5176869	0.3418641	93,453,627.81
AhTCP17	AhTCP49	0.004927137	0.0184398	0.2672016	1,135,454.02
AhTCP18	AhTCP48	0.002091869	0.0270387	0.0773659	1,664,941.75
AhTCP19	AhTCP47	0.006796026	0.0295969	0.2296192	1,822,471.38
AhTCP20	AhTCP25	0.399074138	2.0537667	0.1943133	126,463,464.47
AhTCP21	AhTCP36	0.31837927	1.0005152	0.3182153	61,608,078.22
AhTCP21	AhTCP40	0.328553957	1.0062286	0.3265202	61,959,888.76
AhTCP21	AhTCP50	0.031473149	0.0397444	0.7918890	2,447,315.12
AhTCP22	AhTCP51	0.008109557	0.0587456	0.1380453	3,617,340.68
AhTCP36	AhTCP40	0.02250689	0.0348983	0.6449280	2,148,909.87
AhTCP37	AhTCP39	0.005602267	0.0242796	0.2307399	1,495,047.83
AhTCP36	AhTCP50	0.288229	0.8752547	0.3293087	53,894,994.64
AhTCP40	AhTCP50	0.309598196	1.0016415	0.3090908	61,677,433.92
AhTCP41	AhTCP48	0.416790346	1.1703064	0.3561378	72,063,199.75

## Data Availability

The datasets featured in this study are accessible within the online PGR database (http://peanutgr.fafu.edu.cn/; accessed on 25 July 2024). Furthermore, upon reasonable request, the datasets utilized and/or analyzed throughout this research can be obtained from the corresponding author. Nonetheless, the majority of the data have been included in the Appendix A.

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
