# Peer review of "Genome-Wide Identification and Expression Analysis of TCP Transcription Factors Responding to Multiple Stresses in Arachis hypogaea L."

_ijms, 2025, doi:10.3390/ijms26031069_

Round 1
Reviewer 1 Report
Comments and Suggestions for Authors
Given that the current manuscript we are reading, after careful study, it can be found that its research content is currently only limited to the bioinformatics analysis level. However, considering the in-depth and comprehensive requirements of scientific research, it inevitably leads us to such a question: Can merely relying on the existing bioinformatics analysis be sufficient to fully reveal the complex biological mechanisms contained in the differentially expressed genes? From a more rigorous scientific research perspective, if the research method of functional assays is introduced, will it be possible to unearth more key information hidden behind these differentially expressed genes? The minor problems that occur in this manuscript:
1. During the discussion process, it is necessary to ensure that all species names are uniformly presented in italic format, so as to guarantee the standardization and accuracy of term expression.
2. In section 4.4 of the "Methods" chapter of this manuscript, remove all the font formats that are set to bold and black, and change them to the regular font style.
Comments on the Quality of English LanguageThis manuscript involves a lot of technical terms, so it needs to be modified by professionals.
Author Response
Given that the current manuscript we are reading, after careful study, it can be found that its research content is currently only limited to the bioinformatics analysis level. However, considering the in-depth and comprehensive requirements of scientific research, it inevitably leads us to such a question: Can merely relying on the existing bioinformatics analysis be sufficient to fully reveal the complex biological mechanisms contained in the differentially expressed genes? From a more rigorous scientific research perspective, if the research method of functional assays is introduced, will it be possible to unearth more key information hidden behind these differentially expressed genes?
Re:Thank you very much for your suggestion. Yes, the current manuscript is mainly based on the bioinformatics analysis to characterize the AhTCP genes at the whole genome level. Then the expression profiles of AhTCPs in different tissues and responsing to different stresses were anayzed. Furthermore, qRT-PCR analysis was perfomed to calidate the accuracy and reliability of the RNA-seq results. The complex biological mechanisms need more rigorous experimental design and more detailed and careful experimental research. Here one of the main, but not the only, aims of our study was to supply AhTCP genes resources in detail for researchers, which is beneficial for further revealing the AhTCP biological function in peanut growth and development.
The minor problems that occur in this manuscript:
- During the discussion process, it is necessary to ensure that all species names are uniformly presented in italic format, so as to guarantee the standardization and accuracy of term expression.
Re:Thank you very much for your careful review. We have check all species names in the manuscript and modified them in italic format.
In section 4.4 of the "Methods" chapter of this manuscript, remove all the font formats that are set to bold and black, and change them to the regular font style.
Re:Thank you very much for your careful review. We have made a rivision.
Reviewer 2 Report
Comments and Suggestions for Authors
The manuscript: Genome-wide identification and expression analysis of TCP transcription factors responsing to multipe stresses in Arachis hypogaea L. by YanTing et al., is an in silico work without any self experimental data being presented by the authors.
The manuscript is in need of substantial English improvement and writing editing, but duo to the missing line numbers I am not able to be more specific.
Several spaces should be added or removed. The references are not prepared according to the journal standard.
The introduction is missing a logical flow of information and is full of several isolated sentences that are difficult to comprehend.
The images are of low resolution and difficult to grasp by the readers.
These are just some minor issues and although the presented analyzes and bioinformatics could be useful for the future research, but without substantiating these data with experiments, the manuscript is not enough to be published. It is common that authors, who present experimental results, use such bioinformatics approach beforehand anyway to draw their hypothesis (but they rather do it themselves).
Therefore, I suggest the authors to design small experiments including at least 2 different abiotic stresses and look at the expression of the selected TCP transcription factors and even try to interfere with their expression. If such an experimental data can be supplemented, then this manuscript can be considered as an important and reliable work to add to the existing knowledge.
Comments on the Quality of English LanguageThe English could be improved to more clearly express the research.
Author Response
The manuscript: Genome-wide identification and expression analysis of TCP transcription factors responsing to multipe stresses in Arachis hypogaea L. by YanTing et al., is an in silico work without any self experimental data being presented by the authors.
Re:Thank you very much for your careful review. Yes, we have performed the qRT-PCR to validate the certain AhTCPs expression changes under different stresses inculding cold, SA and ABA treatments.
The manuscript is in need of substantial English improvement and writing editing, but duo to the missing line numbers I am not able to be more specific.
Re:Thank you very much for your careful review. We have made a rivision and editing to improve our manuscript. We also add the line numbers in the manuscript.
Several spaces should be added or removed. The references are not prepared according to the journal standard.
Re:Thank you very much for your careful review. Yes, we have made careful revisions and the references were also modified to satisfy the journal standard.
The introduction is missing a logical flow of information and is full of several isolated sentences that are difficult to comprehend.
Re:Thank you very much for your careful review. Yes, we have made careful revisions for the introduction and make it logical.
The images are of low resolution and difficult to grasp by the readers.
Re:Thank you very much for your careful review. Yes, we have updated the images with high resolution in the article.
These are just some minor issues and although the presented analyzes and bioinformatics could be useful for the future research, but without substantiating these data with experiments, the manuscript is not enough to be published. It is common that authors, who present experimental results, use such bioinformatics approach beforehand anyway to draw their hypothesis (but they rather do it themselves).
Therefore, I suggest the authors to design small experiments including at least 2 different abiotic stresses and look at the expression of the selected TCP transcription factors and even try to interfere with their expression. If such an experimental data can be supplemented, then this manuscript can be considered as an important and reliable work to add to the existing knowledge.
Re:Thank you very much for your suggestion. Yes, we have performed the qRT-PCR to validate the certain AhTCPs expression changes under different stresses inculding cold, SA and ABA treatments to further confirm the accuracy and reliability of the transcriptome expression data.
Round 2
Reviewer 1 Report
Comments and Suggestions for Authors
Accept.
Author Response
Thank you very much for your careful review.
Reviewer 2 Report
Comments and Suggestions for Authors
The manuscript is now supplemented with new data. Although it is unclear why different genes were selected for different treatments! I suggest writing some lines about this issue even if it is related to technical circumstances.
Line 237: change ‘’2.7. Expression analysis of peanut TCP gene family’’ to: ‘2.7. In silico expression analysis of peanut TCP gene family‘
Line 242: revise: ‘’…in different in different tissues…’’
The spacing issue is still there, please correct ‘’ ethephon(Figure 6b)’’ and similar incidents including the spaces before all of the [s.
Line 291: change ‘’2.8. Expression of AhTCPs in real-time under treatments with cold, ABA, and SA’’ to: ‘2.8. Expression of selected AhTCPs genes under cold, ABA, and SA treatments’
Line 297: ‘’AhTCPs exhibited corresponding response patterns’’ ! be more specific!
Line 298 and elsewhere: italicize the name of the genes: AhTCP33…
Line 306: ‘’…transcriptome data’’ Here you can refer to a relevant figure: ‘…transcriptome data (Figure x)’’
Line 316: change: ‘’Real-time expression analysis of AhTCP genes…’’ to ‘Relative expression analysis of AhTCP genes in peanut seedlings (leaf or root or whatever tissue it was) in response to…’
Line 318: ‘’…and statistical significance was ascertained through ANOVA.’’ I do not see any statistics on the charts!!!
Line 341’’: …which was similar to the results of previous studies on TCP transcription factors’’ add some references here after the end of the sentence!!!
Line 347: here and everywhere in the dissection you should refer to relevant results by mentioning the figures: e.g. ‘’…Class II CYC/TB1 proteins’’ (Figure x)’. Also bring relevant references after every general statements.
Line 372: change: ‘’…remains further studies.’’ to ‘requires further studies’.
Line 506: 4.7. Stress treatments and qRT-PCR analysis: Here you need give more information to make your trial reproducible for others. Give more details such as culture condition, basic fertigation intervals, compositions, and quantity, growth medium type and character. The quantity of the applied ABA and SA with the given concentrations, the duration of the applied cold stress, number technical and biological repeats, the tissues that were subjected for RNA extraction, the primers list and characters and their origin (accession numbers), the instrument that was used, the PCR efficiency for every pair of primers. Just imaging that one should read this part and be able to recreate the experimental condition without any questions being unanswered!
You are also missing the quantification of the applied phytohormones in plant tissues. If you can also provide these data, this work will be more complete! I also suggest including a few images of the plants during the experiments. It would be very informative to see at least some images to grasp the impact of the applied treatments on peanuts seedlings morphology!
If you applied any statistival analysis as stated in Fig 7 caption, then bring a subsection and fully describe the applied methods and P values, normality tests or any methods that may have been used for homogeny tests.
Line 523: The conclusion is very weak, try to expand it and say the most important facts that you want to communicate!! Especially the results that have been confirm based on RNA-seq and qRT-PCR data. Mention the main biological functions that are regulated by these genes.
Author Response
The manuscript is now supplemented with new data. Although it is unclear why different genes were selected for different treatments! I suggest writing some lines about this issue even if it is related to technical circumstances.
Re: Thank you for your suggestions. Based on the transcriptome data, we have selected genes that exhibited significant upregulation or downregulation in response to low temperature, SA, and ABA stress for further qRT-PCR analysis. We have included these additional details in our article.
Line 237: change ‘2.7. Expression analysis of peanut TCP gene family’ to: ‘2.7. In silico expression analysis of peanut TCP gene family‘
Re: Thank you for your suggestions. We have already made the modifications in the manuscript.
Line 242: revise: ‘…in different in different tissues…’
Re: Thank you very much for your careful review. We have already made the modifications in the manuscript.
The spacing issue is still there, please correct ‘ethephon(Figure 6b)’and similar incidents including the spaces before all of the [s.
Re: Thank you very much for your careful review. We have addressed all issues related to spacing in the manuscript.
Line 291: change ‘2.8. Expression of AhTCPs in real-time under treatments with cold, ABA, and SA’ to: ‘2.8. Expression of selected AhTCPs genes under cold, ABA, and SA treatments’
Re: Thank you for your suggestions. We have already made the modifications in the manuscript.
Line 297: ‘AhTCPs exhibited corresponding response patterns’ ! be more specific!
Re: Thank you for your suggestions. We have already made the modifications in the manuscript.
Line 298 and elsewhere: italicize the name of the genes: AhTCP33…
Re: Thank you very much for your careful review. We have italicized the gene names throughout the manuscript.
Line 306: ‘…transcriptome data’ Here you can refer to a relevant figure: ‘…transcriptome data (Figure x)’’
Re: Thank you for your suggestions. We have made the necessary additions to the manuscript.
Line 316: change: ‘Real-time expression analysis of AhTCP genes…’ to ‘Relative expression analysis of AhTCP genes in peanut seedlings (leaf or root or whatever tissue it was) in response to…’
Re:Thank you for your suggestions. We have already made the modifications in the manuscript.
Line 318: ‘…and statistical significance was ascertained through ANOVA.’ I do not see any statistics on the charts!!!
Re: Thank you very much for your careful review. We have already added the results of the statistical analysis to the charts.
Line 341’’: …which was similar to the results of previous studies on TCP transcription factors’’ add some references here after the end of the sentence!!!
Re:Thank you for your suggestions. We have included relevant references in the manuscript.
Line 347: here and everywhere in the dissection you should refer to relevant results by mentioning the figures: e.g. ‘’…Class II CYC/TB1 proteins’’ (Figure x)’. Also bring relevant references after every general statements.
Re:Thank you for your suggestions. We have incorporated the corrections into the relevant sections of the manuscript.
Line 372: change: ‘’…remains further studies.’’ to ‘requires further studies’.
Re:Thank you for your suggestions. We have already made the modifications in the article.
Line 506: 4.7. Stress treatments and qRT-PCR analysis: Here you need give more information to make your trial reproducible for others. Give more details such as culture condition, basic fertigation intervals, compositions, and quantity, growth medium type and character. The quantity of the applied ABA and SA with the given concentrations, the duration of the applied cold stress, number technical and biological repeats, the tissues that were subjected for RNA extraction, the primers list and characters and their origin (accession numbers), the instrument that was used, the PCR efficiency for every pair of primers. Just imaging that one should read this part and be able to recreate the experimental condition without any questions being unanswered!
Re:Thank you for your suggestions. We have added relevant content in the section titled "4.7. Stress Treatments and qRT-PCR Analysis."
You are also missing the quantification of the applied phytohormones in plant tissues. If you can also provide these data, this work will be more complete! I also suggest including a few images of the plants during the experiments. It would be very informative to see at least some images to grasp the impact of the applied treatments on peanuts seedlings morphology!
Re: Thank you for your suggestions. The qRT-PCR analysis conducted in this study serves to verify the accuracy and reliability of the RNA-seq results. In the laboratory, transcriptome sequencing was performed on peanut leaves after stress treatment was applied to the peanuts. Therefore, conducting qRT-PCR analysis solely on peanut leaves is sufficient to demonstrate the accuracy and reliability of the RNA-seq results. After spraying the peanut leaves with SA and ABA for 24 hours, there were no obvious changes in the morphology of the peanut plants. Following a 24-hour low-temperature treatment on the peanut plants, only slight wilting was observed in the peanut leaves, which is why no photographs were taken.
If you applied any statistival analysis as stated in Fig 7 caption, then bring a subsection and fully describe the applied methods and P values, normality tests or any methods that may have been used for homogeny tests.
Re: Thank you for your suggestions. We have already included the supplements in the article.
Line 523: The conclusion is very weak, try to expand it and say the most important facts that you want to communicate!! Especially the results that have been confirm based on RNA-seq and qRT-PCR data. Mention the main biological functions that are regulated by these genes.
Re:Thank you for your suggestions. We have added relevant content to the conclusion section.
Round 3
Reviewer 2 Report
Comments and Suggestions for Authors
I do not see the Table S1 and S2 in the Supp. Mat.
Please also give a reference for the origin of the primers. An accession number of the genes in a database is recommended. If you designed the primers, you can mention it with the name of the applied software.
Line 302: ''(Figure 7a) .'' remove the space after '').'' Do another final check for this kind of errors.
Between the reference numbers you do not need space after ,: [37,44], but between the Figures name you need: (Figure 6b, 6c).
Line 528: I think it is Actin and not ''Ahactin''. Also add the primer sequence of actin to table S3.
Line 532-533: ''The GraphPad Prism 7.0 software was employed to compare differences between the control and experimental value''. Give the name of the applied method (e.g. Duncan's multiple-range test or Tukey's). The software name is not enough.
Author Response
I do not see the Table S1 and S2 in the Supp. Mat.
Re: Thank you very much for your careful review. We have already uploaded Table S1 and Table S2.
Please also give a reference for the origin of the primers. An accession number of the genes in a database is recommended. If you designed the primers, you can mention it with the name of the applied software.
Re: Thank you for your suggestions. We designed the primers using BioXM 2.7.1 and have included this information in the article.
Line 302: ''(Figure 7a) .'' remove the space after '').'' Do another final check for this kind of errors.
Re: Thank you very much for your careful review. We have already made the necessary modifications in the article and carefully checked for such errors.
Between the reference numbers you do not need space after ,: [37,44], but between the Figures name you need: (Figure 6b, 6c).
Re: Thank you very much for your careful review. We have already made the modifications in the article.
Line 528: I think it is Actin and not ''Ahactin''. Also add the primer sequence of actin to table S3.
Re: Thank you for your suggestions. We have made modifications in the article and have also added the primer sequences.
Line 532-533: ''The GraphPad Prism 7.0 software was employed to compare differences between the control and experimental value''. Give the name of the applied method (e.g. Duncan's multiple-range test or Tukey's). The software name is not enough.
Re: Thank you for your suggestions. We conducted significance analysis using ANOVA in GraphPad Prism 7.0. We have already included this information in the article.